# Importance and Reality of TDM for Antibiotics Not Covered by Insurance in Japan

**DOI:** 10.3390/ijerph19052516

**Published:** 2022-02-22

**Authors:** Fumiya Ebihara, Yukihiro Hamada, Hideo Kato, Takumi Maruyama, Toshimi Kimura

**Affiliations:** 1Department of Pharmacy, Tokyo Women’s Medical University Hospital, Tokyo 162-8666, Japan; ebihara.fumiya@twmu.ac.jp (F.E.); maruyama.takumi@twmu.ac.jp (T.M.); kimura.toshimi@twmu.ac.jp (T.K.); 2Department of Pharmacy, Mie University Hospital, Mie 514-8507, Japan; katou.hideo.233@mail.aichi-med-u.ac.jp

**Keywords:** HPLC, TDM, ceftriaxone, daptomycin, linezolid

## Abstract

Under the Japanese health insurance system, medicines undergoing therapeutic drug monitoring (TDM) can be billed for medical fees if they meet the specified requirements. In Japan, TDM of vancomycin, teicoplanin, aminoglycosides, and voriconazole, which are used for the treatment of infectious diseases, is common practice. This means the levels of antibiotics are measured in-house using chromatography or other methods. In some facilities, the blood and/or tissue concentrations of other non-TDM drugs are measured by HPLC and are applied to treatment, which is necessary for personalized medicine. This review describes personalized medicine based on the use of chromatography as a result of the current situation in Japan.

## 1. Introduction

In Japan, insurance claims based on therapeutic drug monitoring (TDM) became possible for lithium carbonate in the treatment of manic depression in 1980, followed by antiepileptic drugs and digitalis the following year. Since then, the therapeutic benefits of TDM have been confirmed, the number of drugs covered has gradually expanded, and insurance billing rates have increased. When the blood concentration of the administered drug is measured and the dosage is precisely controlled based on the results, the associated costs can be calculated and billed only once in a calendar month.

The insurance billing fee includes the costs for measuring the blood drug concentration, drawing blood for this measurement, and administering of the dosage based on the results, and the cost of measuring the blood concentration more than once in a single month cannot be calculated separately. In addition, the blood concentration of the drug and the main points of the treatment plan should be detailed in the medical record.

In the 1980s and 1990s, high-performance liquid chromatography (HPLC) played a significant role in the analysis of drug concentrations in blood. However, with the spread of simple automated analyzers based on ligand-binding assays and the promotion of outsourcing to clinical laboratories, the number of medical institutions with HPLC capability has decreased [1]. In the area of clinical toxicology, precision analytical equipment was installed in emergency departments in 1998 with support from the Ministry of Health, Labor, and Welfare, but most of the equipment are systems for high-performance liquid or gas chromatography coupled with mass spectrometer detectors; not many facilities use HPLC with UV–visible detectors as their main equipment [2,3].

Focusing on drugs for infectious diseases, the only drugs that can be billed to insurance in Japan are vancomycin, teicoplanin, aminoglycoside antibiotics, and voriconazole. In some facilities, the blood and tissue concentrations of other drugs are measured by HPLC and are applied to treatment, which is necessary for personalized medicine. In this paper, we focus on antibiotics that are not covered by insurance and introduce the clinical significance of TDM and methods for measuring blood concentrations with examples from our own experience. In this review, we discussed ceftriaxone (CTRX), whose blood and tissue concentrations were actually measured at our hospital, and daptomycin (DAP), linezolid (LZD), and tedizolid (TZD), which are listed in the Japanese Antimicrobial Agents TDM Guidelines 2022 [4].

## 2. TDM of Ceftriaxone

### 2.1. Characteristics of Ceftriaxone and Significance of TDM

CTRX is a third-generation cephalosporin that has a broad antibacterial spectrum, good tissue migration, and a longer half-life than other cephalosporins, allowing for once-daily administration. CTRX has been used for a variety of indications and has many opportunities for use [5]. In addition, dose adjustment is not necessary for patients with impaired renal function [6]. Patel et al. reported that adjusting the dosage regimen of ceftriaxone for patients with impaired renal function should not be necessary when the daily dosage is 2 g or lower [7]. Antibiotic-associated encephalopathy (AAE) is a known side effect of cephalosporins. AAE is classified into three major categories: type I for cephalosporins and penicillins, type II for quinolones and macrolides, and type III for metronidazole; type I is said to be free of epileptic waves [8]. In particular, there have been many reports on AAE with cefepime (CFPM), a fourth-generation cephalosporin drug, including reports on the intoxication zone of blood levels responsible for AAE [9,10,11]. There are also scattered reports of AAE caused by CTRX, and according to an adverse drug reaction database study in France, the serious CNS adverse effects caused by CTRX and CFPM were reported to be similar [12]. AAE with CTRX has been reported more frequently in the elderly and patients with chronic kidney disease and is more likely to occur when high doses are given and blood and cerebrospinal fluid (CSF) concentrations are high [13,14,15].

In a recent report, Lacroix et al. [16] used the French pharmacovigilance database to analyze records of CNS adverse events due to CTRX that were recorded as occurring in 1995–2017. A total of 152 serious adverse drug reactions (ADRs) were analyzed: 112 resulted in hospitalization or prolonged hospitalization (73.7%), 12 resulted in death (7.9%), and 16 were life-threatening. The median age was 74.5 years, and the median rate of creatinine clearance (CLcr) was 35 mL/min. The median time of onset was 4 days; the mean daily dose was 1.7 g, with three patients receiving doses exceeding the maximum recommended dose; and plasma ceftriaxone levels were recorded in 19 patients (12.5%), with eight patients exceeding the threshold for toxicity (>100 μg/mL). In addition, electroencephalography performed on 50 patients (32.9%) showed abnormalities in 37 (74%).

Although the frequency of CTRX-related encephalopathy in Japan is not clear, it has been reported that the sales volume of third-generation cephalosporins is about six times higher than that of fourth-generation cephalosporins [17], and caution is required.

### 2.2. Report on the Measurement of Blood Levels and CSF of CTRX in Japan

In Japan, we searched for clinically applied studies on CTRX that are not generally clinically measured. As a result, we present reports on the measurement of CTRX concentrations in cerebrospinal fluid by HPLC−UV in Japan.

Kotani et al. [18] measured CTRX concentrations in CSF samples collected from peritoneal dialysis patients diagnosed with CTRX-induced encephalopathy (CIE) using an HPLC–UV system to investigate whether high CTRX concentrations in CSF are associated with CIE. Their study used an octadecyl silica (ODS) column, methanol, and a mobile phase of mixed solution (25:75, *v*/*v*) in 10 mM phosphate to accurately analyze the CSF samples from CIE patients; the detection wavelength was 280 nm. Based on the current HPLC–UV capability, a linear range of 0.1–100 μg/mL (r = 0.999) was obtained. In a recovery study using a blank sample of human CSF and a control serum supplemented with CTRX, the recovery of CTRX was ≥95.3%, and the RSD was <5.8% (*n* = 3). CTRX in CSF and serum obtained from a patient diagnosed with CIE was measured using the developed HPLC–UV system. The concentration of CTRX in CSF and serum was 2.61 and 37.35 μg/mL, respectively. We simulated the serum and CSF concentrations of CTRX using simulation software developed by Oda (Figure 1) [19].

Suzuki et al. [20] reported the case of a patient with renal failure who experienced encephalopathy induced by CTRX. An 86-year-old woman undergoing maintenance hemodialysis was treated with CTRX for *Helicobacter cinaedi* bacteremia, and her mental status worsened during antibiotic administration. During the period when CTRX was administered at 2 g/day, the measured plasma and CSF CTRX concentrations were high (>100 and 10.2 μg/mL, respectively). The patient reported that her mental status improved after the antibiotic treatment was stopped. Table 1 shows the PubMed search results for reports that measured blood and CSF concentrations of CTRX, DAP, LZD, and TZD in Japan.

In Japan, the measurement of the CTRX blood concentration is not covered by insurance and is limited to facilities with appropriate laboratory equipment. In practice, there is a time lag between collecting samples and obtaining blood concentration results, making it difficult to use them for differential diagnosis in real time. LC–MS systems can measure blood levels more accurately than HPLC systems, but are expensive and therefore impractical. In the future, it would be desirable to have a test system that can be used in general practice so that the CTRX blood concentration can be measured when AAE is suspected. These reports suggest that TDM of CTRX may be able to prevent the occurrence of AAE.

## 3. TDM of Daptomycin

### 3.1. Characteristics of Daptomycin and Significance of Blood Level Measurement

DAP is a lipopeptide antibiotic that is effective against antibiotic-resistant Gram-positive bacteria such as methicillin-resistant *Staphylococcus aureus* (MRSA) and vancomycin-resistant enterococci (VRE) [34,35]. DAP has a molecular weight of about 1.6 kDa, a high protein-binding rate (90–95%), and a distribution volume of 0.1 L/kg [22]. The elimination half-life of DAP is approximately 8 to 9 h in adults [36]. Currently, TDM of DAP is not routinely performed in clinical practice in Japan. A once-daily dose of 4 mg/kg is administered for skin and soft tissue infections and 6 mg/kg for sepsis and infective endocarditis of the right heart system [37]. DAP has also been administered every 48 h in patients whose CLcr is less than 30 mL/min or for those who require dialysis [38]. As this drug is mainly excreted by the kidneys, the dosage should be adjusted for patients with impaired renal function.

It has been reported that the efficacy of DAP is strongly correlated with the area-under-the-curve/minimum inhibitory concentration (AUC/MIC) ratio and peak concentration/MIC (Cpeak/MIC) ratio [39], and it is considered to be drug concentration dependent. In clinical studies, an AUC/MIC of 666 or higher in MRSA infections was associated with a lower mortality [40], and a trough concentration of less than 3.18 μg/mL (steady state) was associated with poorer clinical outcomes [41]. A typical side effect of DAP is increased creatine phosphokinase (CPK) levels [42,43]. In particular, it is known that a blood level of 24.3 μg/mL or higher increases the risk of higher CPK levels [42]. On the other hand, safety and tolerability at high doses (≥8 mg/kg) have also been reported [44,45,46], and the correlation between elevated CPK levels and dosage and blood levels is not clear. Blood levels of DAP vary widely from patient to patient, and factors that may contribute to this variability include renal function, hemodialysis, continuous renal replacement therapy, obesity, hypoalbuminemia, and the pathogenesis of severe infections [41,47,48,49,50]. At this stage, there is insufficient evidence to conclusively determine a correlation between the efficacy and adverse effects of DAP and blood levels of the drug, and it is unclear whether TDM of DAP should be recommended for clinical practice in Japan.

### 3.2. Report on the Measurement of Blood Levels of Daptomycin in Japan

We present a report on the measurement of blood levels of DAP in clinical practice in Japan, and its application to the treatment of infectious diseases.

Urakami et al. [24] used Monte Carlo simulation and TDM to investigate the best way to manage DAP based on PK/PD parameters. Serum concentrations of DAP in 16 MRSA-infected patients were measured with the HPLC–UV system. First, venous blood (5 mL) was collected, and the blood sample was centrifuged at 5000× *g* for 10 min and stored at −30 °C until plasma analysis. The lower limit of quantification for this assay was 0.78 μg/mL. The analysis column was a TSK gel Octyl −80 Ts, 5 μ, 250 × 4.6 mm (TOSOH, Tokyo, Japan), UV wavelength was 214 nm, and the mobile phase was 40 mM phosphate ammonium buffer (pH 4.5)/acetonitrile = 60:40 *v*/*v*. All of the solvents used were HPLC grade [24].

As a DAP pharmacokinetics parameter, the volume of distribution of patients in the study was larger than the volume of distribution among healthy Japanese subjects. The half-life of this drug was 8.9 to 34.9 h, which gradually increased as CLcr decreased. In the Monte Carlo simulation, the cumulative fraction of response (CFR) for Cpeak/MIC ≥ 60 [39] and AUC/MIC ≥ 666 at 6 mg/kg every 24 h was 72.0% and 78.8%, respectively, whereas at 10 mg/kg every 24 h, both CFR values improved to 99%. With TDM of DAP at 6 mg/kg every 24 h, the target peak and AUC were reached in 40% of patients (2 of 5). In that study, they reported that TDM is necessary because of individual differences in PK with DAP. A high-dose regimen of 8 mg/kg or higher may be required to ensure efficacy, especially in Japanese patients with normal renal function. In this study, one patient with a trough level of 49.4 μg/mL and CLcr of 22.4 mL/min had elevated CPK, but no other patients had adverse events attributable to DAP.

Yamada et al. [25] investigated the relationship between DAP trough concentration (Cmin) and CPK elevation to determine the optimal DAP administration. DAP concentrations in the plasma of 20 patients were measured. Plasma samples were collected at trough and Cpeak within 60 min after the end of infusion on day 3 after DAP administration. HPLC analysis was coupled with use of a UV detector set to a detection wavelength of 214 nm. The column used was Octyl-80Ts (4.6 × 250 mm), temperature was 37 °C, the mobile phase was acetonitrile/ammonium phosphate buffer (40 mM, pH 4) (40:60), and flow rate was 1.5 mL/min. The lower limit of quantification for HPLC was 1.0 μg/mL, and the intra- and inter-day coefficients of variation were less than 5.0%. Logistic regression analysis was performed, and the Cmin of DAP was significantly associated with elevated CPK and was concentration dependent (odds ratio 1.21, *p* = 0.048). Patients with DAP Cmin < 19.5 μg/mL did not show increased CPK, but those with Cmin > 19.5 μg/mL had a high rate of increased CPK (4/5, 80%), and three of these patients showed increased CPK after one week of treatment. Based on the results of the Monte Carlo simulation to determine the optimal dose of DAP, the estimated doses were 4–6 mg/kg/day when the MIC was 0.5 μg/mL or less and 10 mg/kg/day when the MIC was 1 μg/mL.

In addition to the above reports, the use of liquid chromatography–tandem mass spectrometry (LC–MS/MS) [26], ultra-performance liquid chromatography coupled to tandem mass spectrometry (UPLC–MS/MS) [27], and HPLC equipped with a photodiode array (UHPLC–PDA) [28] was also studied.

Based on the above reports, the clinical significance of TDM of DAP is considered to be significant, especially in patients with impaired renal function, hemodialysis, continuous renal replacement therapy, and obesity, because of the large variation in blood levels among patients. It is also expected to avoid the risk of elevated CPK and the emergence of resistant bacteria. However, there are limited data to support the use of TDM in DAP.

## 4. TDM of Linezolid

### 4.1. Characteristics of Linezolid and Significance of Blood Level Measurement

LZD is an oxazolidinone drug that has an excellent antibacterial activity against Gram-positive bacteria, including MRSA and VRE [51,52]. The plasma protein-binding rate and volume of LZD distribution in adults are 31% and 40–50 L, respectively [53]. Myelosuppression, including anemia and thrombocytopenia, has been reported as a serious side effect of LZD, and is generally reversible when discontinuing treatment, with recovery usually taking 1–2 weeks [54]. LZD does not require dosage adjustment with or without renal dysfunction [55,56]. The PK/PD parameter is the percentage of time that the plasma concentration exceeds the MIC (% T > MIC) by more than 85%, and the AUC/MIC ratio is 80–120 [57,58]. In terms of thrombocytopenia, the recommended range of trough concentrations for LZD is 2.0–7.0 μg/mL [59]. Patients with impaired renal function may have significantly lower platelet counts than those with normal renal function, and there are reports of therapeutic outcomes with TDM of LZD [60,61,62].

### 4.2. Report on the Measurement of Blood Levels of Linezolid in Japan

There are case reports of successful treatment of infection when Japanese hospital pharmacists performed TDM of LZD.

Tsuji et al. [29] investigated mediastinitis after cardiac surgery that was caused by MRSA in a patient with renal dysfunction by measuring trough concentrations in serum and wound exudate to adjust the LZD dose. As the patient’s serum creatinine level and glomerular filtration rate were 5.6 mg/dL and 8.6 mL/min/1.73 m^2^, respectively, LZD was administered at a single dose of 600 mg daily. On the 21st day of therapy, the serum Cmin was 11.5 μg/mL and the platelet count decreased to 65,000/μL. The cessation of LZD administration for one day made the Cmin decrease to 3.5 μg/mL on the 23rd day. Although linezolid therapy restarted on the day 24th day, the Cmin increased to 9.3 μg/mL on the 27th day. Thus, the patient was administered a single dose of 300 mg daily. As a result, the Cmin was maintained between 2 and 7 μg/mL and the platelet count recovered. Moreover, there was little decrease in efficacy with the change in dosage.

Next, Matsuda et al. [30] reported that two patients were cured with good control of the platelet count through the adjustment of LZD dosage by TDM. One patient who diagnosed pyogenic spondylitis caused by MRSA was administered 600 mg twice daily. The patient’s serum creatinine level was 1.37 mg/dL. The platelet count started to decrease from the next day of therapy and to 94,000/μL on the 9th day of therapy. The Cmin was 39.4 μg/mL, and therefore the patient was administered a single dose of 600 mg daily. On the next day of the dose reduction, the Cmin was 35.1 μg/mL, and the platelet count continued to decrease to 65,000/μL. Fortunately, as MRSA was not detected from abscess culture, LZD administration was discontinued. However, the platelet reduction continued for three days after discontinuing LZD administration and the nadir of platelet count was 28,000/μL. Another who had MRSA detected from the from abscess culture was administered 600 mg twice daily. The patient’s serum creatinine level was 0.75 mg/dL. The Cmin was 22.3 μg/mL on the 2nd day of therapy, and then the dosage of LZD was decreased to 600 mg once daily. The patient completed the treatment of LZD for 28 days without thrombocytopenia. The measurement of Cmin early after LZD administration could lead long-term administration with tolerance for thrombocytopenia.

Ashizawa et al. [31] performed TDM of LZD for patients treated with a combination of LZD and rifampicin (RFP) for osteomyelitis caused by MRSA. The concomitant use of LZD and RFP may decrease the serum concentration of linezolid due to drug interactions, which may reduce the therapeutic effect [59]. The patient was treated with a dose of LZD 600 mg twice daily and a dose of RFP 450 mg once daily, and the Cmin was maintained within an optimal range of 3.7–7.3 μg/mL. They concluded that TDM could be useful for keeping the efficacy and safety of combination therapy with LZD and RFP.

These reports suggest that TDM should be considered in terms of tolerability, especially in conditions that require long-term administration of LZD, such as osteomyelitis and meningitis of MRSA.

## 5. TDM of Tedizolid

### 5.1. Characteristics of Tedizolid and Significance of Blood Level Measurement

TZD phosphate is a novel oxazolidinone pro-drug that is converted to TZD by endogenous phosphatases [63,64]. TZD has s microbiologic activity against a broad range of Gram-positive pathogens, including resistant strains such as MRSA and VRE [65,66]. The profile of TDZ is roughly similar to LZD, which is the first commercialized oxazolidinone antibiotic. The dose of TZD does not need to be modified in patients with renal impairment or on hemodialysis. Due to the absorptive bioavailability with over 80%, TZD is administered both orally and intravenously [67,68], the time to maximum concentration is achieved within approximately 3 h of oral dosing, and steady-state plasma concentrations are reached within 3 days of the initial daily dose [69,70]. Moreover, TZD shows moderately protein bound with over 80% in human plasma and is well distributed into tissue where unbound concentrations are almost equal to the free concentrations in the plasma [71]. The PK parameters and penetration of adipose tissue and muscle of TZD following the single oral administration of 200 mg and pulmonary of TZD following the oral administration of 200 mg once daily are shown in Table 2. The mean AUC_0–12_ values for adipose tissue and muscle were 5.3 and 5.9 mg h/L, respectively. The mean rates of penetration into the adipose tissue and muscle using the AUC_0–12_ in the plasma were 1.1 and 1.2, respectively [70]. The estimated AUC_0–24_ value for the pulmonary was 106.0 mg h/L. The rate of the AUC_0–24_ in the pulmonary over the AUC_0–24_ in the plasma was 39.7 [71].

On the other hand, administration is once per day as TZD shows a long elimination half-life of 12 h. The PK profile was similar after single and multiple daily doses because of a linear PK [72]. The mean AUC was 30 mg h/L, the mean maximum plasma concentration was 2.6 μg/mL, and the mean elimination half-life was 11 h after a single intravenous 200 mg dose [69]. TZD is primarily metabolized by the liver, and only 18% of the administered drug is eliminated in the urine [73]. The PK/PD index that best correlates with efficacy is the free drug AUC/MIC (fAUC/MIC). The target fAUC/MIC required for antimicrobial efficacy was over 3 [74].

To date, there is no available information on TDM of TZD and dosing optimization. However, it has been reported that the AUC depending on body weight shows individual difference and that thrombocytopenia caused by TZD depends on dose [75,76]. Moreover, a recent report on clinical application of measuring the concentrations showed an approximately eight-fold difference in a trough concentration between patient with a CLcr of 64.5 mL/min and that of 50.2 mL/min [32]. In the future, measuring the TZD concentrations in clinical settings might be needed.

### 5.2. Report on the Measurement of Blood Levels of Tedizolid in Japan

Several methods for the quantification of TZD in plasma using HPLC have been reported [69,70,71,77,78,79,80,81,82,83]. In Japan, three reports were published in 2020–2021.

Kai et al. reported the quantification in plasma samples from three patients in the intensive care unit (ICU) using ultra-high-performance liquid chromatography coupled with tandem mass spectrometry (UHPLC−MS/MS) [33]. The concentration range of calibration curves for TZD was 0.01–5 μg/mL. The measured concentrations in blanks were less than 20% of the peak response of the lower limited of quantification (LLOQ) and less than 5% for internal standard (IS). Blood samples were collected before infusion (Cmin) and 1 h after infusion for TZD (Cpeak). The ranges of Cmin and Cpeak in patients with CLcr of 51.7–60.4 mL/min were 0.06–0.12 and 2.67–4.01 μg/mL. Next, Tanaka et al. reported the quantification in plasma samples using ultra-performance liquid chromatography coupled with MC/MC (UPLC−MS/MS) [32]. The regression coefficients of the actual concentration versus back-calculated concentration was over 0.99 for a range from 0.005 to 5 μg/mL, which showed the linearity. The LLOQ and three quality controls (QCs of low, medium, and high) were less than 15% for both accuracy and precision. Two patients received intravenous infusion of TZD at a dose of 200 mg once daily during the ICU admission. Blood sampling was conducted as Cpeak at 1 h after the initiation of infusion and as Cmin just prior to the infusion of TZD. The measured Cpeak and Cmin of TZD ranged from 1.87 to 4.92 μg/mL and from 0.09 to 0.78 μg/mL, with remarkable individual differences between ICU patients with the CLcr of 50.2 and 64.5 mL/min. It took a short run time of 5 min per sample, which allowed for a relatively rapid feedback of the results to physicians. Considering these individual differences and the promptness of measurement, TDM of TZD would be required for ICU patients as a first target population.

Finally, Tsuji et al. reported the quantification of serum TZD concentrations using high-performance liquid chromatography-fluorescence (HPLC-FL) [84]. High linearity with R^2^ > 0.999 was exhibited over a concentration range from 0.025 to 10 μg/mL for TZD. The range of intra- and inter-assay accuracies of TZD were from 99.2% to 107.0% and from 99.2% to 107.7%, respectively. The range of intra-and inter-assay precisions were from 0.5% to 3.2% and from 0.3% to 4.1%, respectively. The intra- and inter-assay precisions for the LLOQ were 17.0% and 15.3%, respectively. This method has wider ranges of the TZD concentrations than the two studies [32,33]. However, the measurement of TZD concentration in actual patients receiving TZD has not been conducted yet.

At this time, as far as we have been able to determine, we have not found anything definitive on the clinical significance of TDM of TZD.

## 6. The States of TDM for DAP, LZD, and TZD in Other Countries

Several studies have highlighted the importance of TDM to optimize daptomycin use [41]. Nevertheless, there is no recommendation for TDM in any of the current guidelines for daptomycin, and we were not able to find studies reporting the TDM practice in other countries. Recently, a questionnaire survey regarding institutional TDM practice for antibiotics for all Australian hospitals was conducted [85]. TDM for linezolid was less common, with only 5%. Moreover, daptomycin and tedizolid were not even listed in the question. In particular, the role of TDM for tedizolid [86] and daptomycin remains to be proven.

## 7. Limitation

In Japan, to the best of our knowledge, some, but not all, hospitals and clinics have bioanalytical methods validated following the guidelines of the Food and Drug Administration, the European Medicines Agency, and the Japanese regulatory authorities when conducting TDM. This process is very time-consuming. Therefore, it is not practical to validate methods on a limited number of patients. As the number of drugs covered by insurance increases, the need for simpler and faster bioanalytical methods increases. In addition, the validation of standard bioanalytical methods is very costly. In Japan, many facilities cannot perform bioanalytical methods on time, making it challenging to perform TDM of these antibiotics frequently. In Japan, many facilities lack access to the timely bioanalytical method, and most facilities, including universities and laboratories, use research funds to conduct them.

## 8. Conclusions

This paper discusses the need for TDM of antibiotics for which TDM cannot be billed to insurance in Japan, and the reporting of facilities that implement it. TDM, which used to be performed in the laboratory, has shifted to bedside TDM by hospital pharmacists. Therefore, there is a need to build a measurement system that can quickly obtain the results of the drug concentrations. It is expected that pharmacists will be able to use the results of such measurements in real-time at medical sites to help design dosages. As mentioned, some facilities measure the concentration of drugs that cannot be billed for medical treatment at their own facilities, but the number is still tiny. To expand the number of drugs subject to TDM that can be billed to insurance, it is essential to build evidence. In order to build evidence supporting the use of TDM, it is necessary to establish methods for measuring drugs, introduce analytical equipment, and enhance medical staff with expertise. It is desirable to establish a system in which pharmaceutical universities and medical institutions, which are well equipped with these facilities, work together to build evidence for recommending TDM.

## Figures and Tables

**Figure 1 ijerph-19-02516-f001:**
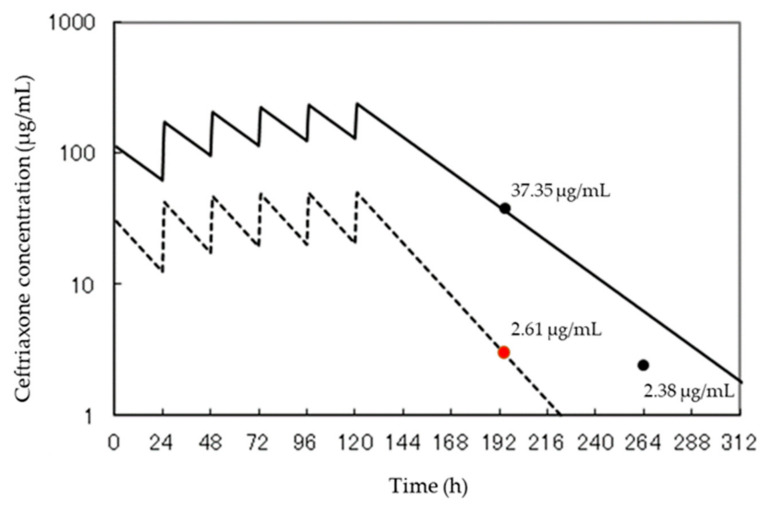
Simulation of blood concentration and human cerebrospinal fluid after ceftriaxone administration [18]. We completed the simulation ourselves using our results. The solid line shows changes in blood concentration after ceftriaxone administration, and the dotted line shows changes in cerebrospinal fluid.

**Table 1 ijerph-19-02516-t001:** Summary of Japanese reports on serum and CSF concentrations of CTRX, DAP, LZD, and TZD [18,20,21,22,23,24,25,26,27,28,29,30,31,32,33].

**Drug**	Characteristics	Objective	Renal Function	Dose	Measurement System	Measurement Accuracy	Blood Concentration	CSF Concentration	Ref.
**CTRX**	Age: 75 Sex: female	Development of HPLC method for accurate, precise, and selective determination of CTRX and its clinical application	peritoneal dialysis	2 g/day	HPLC-UV	-Chromatographic peak heights of CTRX: 0.1–100 μg/mL (r = 0.999) -Detection limit of CTRX: 35 ng/mL -Repeatability (*n* = 6) of the chromatographic peak height for 4.0 μg/mL CTRX: 0.38% RSD.-Recovery rates of CTRX: >95.3%, and these RSDs were <5.8%	37.35 μg/mL	2.61 μg/mL	[18]
	Age: 86 Sex: female	Report of encephalopathy associated with high levels of ceftriaxone in plasma and cerebrospinal fluid, investigation of the causal relationship between ceftriaxone administration and the development of encephalopathy	hemodialysis	2 g/day	HPLC	nd	>100 μg/mL	10.2 μg/mL	[20]
	Population: *n* = 43 patients Sex: male median age: 51.7 years (IQR 33.3–67.1) median BMI: 24.7 kg/m^2^ (IQR 22.4–27.7 kg/m^2^)	Determining the role of transporter genetic variation and blood-brain barrier permeability in predicting ceftriaxone exposure in the central nervous system	estimated creatinine clearance < 30 mL/min	2 g twice a day	HPLC	-Detection limits: 0.24 mg/L in plasma and 0.5 mg/L in CSF -Accuracy: 5.2% for plasma, 7.2% for CSF -Intra- and inter-day coefficients of variation (CV%): 3.6% and 4.5% for plasma samples, and 7.2%, 7.8%, and 10.3% for CSF samples -Recovery rate: 86% (CV% = 3) for CSF samples and 82% (CV% = 8) for plasma samples.	Median Cmax: 157,193.00 ng/mL (IQR 105,164.0–184,852.0 ng/mL)	Median Cmax: 3512.0 ng/mL (IQR 2134.0–6193.0 ng/mL)	[21]
	Population: *n* = 16 patients	Evaluation of tolerability and pharmacokinetic parameters of high-dose ceftriaxone in adult patients treated for central nervous system infections: pharmacological data from two French cohorts	nd	6.5 g/day (range 4–9 g) 97.5 mg/kg (range 77–131 mg/kg)	HPLC	nd	Median total plasma: 69.3 mg/L (range 21.6–201.3 mg/L; *n* =14) Median unbound plasma: 7.95 mg/L (range 0.8–43.7 mg/L; *n* = 8)	Median: 13.3 mg/L (range 0.9–91.2 mg/L)	[22]
	Population: *n* = 7 patients	Investigation of the pharmacokinetics ofboth antibiotics in patients with non- inflammatory obstructive hydrocephalus undergoing external ventricular surgery treated with cefotaxime or ceftriaxone for extracerebral infections	Scr < 1.5 mg/dL	2 g single dose 30 min	HPLC-UV	-Quantification limits of ceftriaxone; 0.8 mg/L in serum and 0.08 mg/L in CSF. -Interday coefficients of variation; 2.0% 249.6; *n* = 6) at 99.7 and 6.8% at 1.55 mg/L inserum and 3.3% at 16.2 and 6.4% at 0.16 mg/L in CSF (*n* = 6).	Cmax: 172.2–271.7 mg/L (median = 249.6; *n* = 6)	Cmax: 0.18–1.04 mg/L (median = 0.43; *n* = 5), confirmed 1–16 h after injection (median = 12 h; *n* = 5).	[23]
**DAP**	Population:16 patients (8 males and 8 females) Age: 70.0 ± 3.4 years weight: 47.6 ± 5.0 kg	Investigate the optimal dosing regimen for daptomycin and determine the need and appropriateness of a high-dose regimen in terms of PK / PD parameters using Monte Carlo Simulation and TDM in a Japanese clinical setting	CLcr 16.2–173.4 mL/min (*n* = 11) hemodialysis (*n* = 5)	single doses (6 mg/kg, 8 mg/kg, 10 mg/kg, and 12 mg/kg) and dosing intervals (24 h and 48 h)	HPLC-UV	-Lowest limit of quantification: 0.78 μg/mL	Cmin: 0.13–49.4 μg/mL Cpeak: 34.2–130.0 μg/mL	nd	[24]
	Population: *n* = 20 patients	Investigate associations between DAP Cmin and creatine phosphokinase elevation via logistic regression analysis (E/R analysis), and to analyze DAP PPK via adaptation of a one-compartment model in Japanese patients to determine optimal DAP doses for minimizing adverse effects and maximizing treatment success by E/R analysis.	CLcr 22.4–213.8 mL/min,	2.8–8.6 mg/kg	HPLC-UV	-Lowest limit of quantitation: 1.0 μg/mL -within-day and between-day coefficients of variation of <5.0%.	Cmin: 2.8–92.4 μg/mL Cpeak: 30.4–76.7 μg/mL	nd	[25]
	Population: *n* = 15 patients	Development of an assay method for the determination of total and free daptomycin in human plasma	nd	4–8 mg/kg once over a 24-hour period.	LC-MS/MS	-Concentration ranges: 1.0–100 μg/mL in total daptomycin and 0.1–10 μg/mL in free daptomycin - Limits of quantitation: 1.0 μg/mL(total daptomycin) and 0.1 μg/mL(free daptomycin) -Recovery rate: total daptomycin measurements ranged from 106.1% and free daptomycin measurements ranged from 98.2%	The plasma concentration ranges of total and free daptomycin in 15 infected patients were 3.01–34.1 and 0.39–3.64 μg/mL	nd	[26]
	Population: two patients admitted to intensive care unit (2 males) Weight: 61.1 kg, 59.0 kg	Development of a new assay for measuring total and free concentrations of daptomycin in plasma with potential clinical applications	CLcr 17.5 mL/min CLcr 140.5 mL/min	-every 48 h of 350 mg (CLcr < 30 mL/min)-once-daily dose of 350 mg (CLcr ≥ 30 mL/min)	UPLC-MS / MS	-Concentration ranges: 0.5–200 μg/mL in total daptomycin and 0.04–40 μg/mL in free daptomycin-Recovery rate: approximately 100% of free daptomycin from ultrafiltration -Limits of quantitation: 0.5 μg/mL (total daptomycin) and 0.04 μg/mL (free daptomycin) -Recovery rate: total daptomycin measurements ranged from 57.1 to 67.4% and free daptomycin measurements ranged from 54.6 to 62.3%	-Patient with low renal function: Cmax of free drug: 2.85 µg/mL (Day 3), 4.2 µg/mL (Day 5) Ctrough of free drug: 0.29 µg/mL (Day 3), 0.86 µg/mL (Day 5) -Patient with normal renal function: Median unbound plasma: Cmax of free drug: 2.69 µg/mL (Day 3), 2.77 µg/mL (Day 5) Ctrough of free drug: 0.77 µg/mL (Day 3), 0.34 µg/mL (Day 5)	nd	[27]
	Population: *n* = 53 patients Sex: Male (*n* = 33), female (*n* = 19)	Examine serum daptomycin levels, creatinine phosphokinase levels, and the incidence of other adverse effects	CLcr ≥ 80 mL/min: *n* = 15 30 ≤ CLcr < 80 mL/min: *n* = 23 CLcr < 30 mL/min: *n* = 14 haemodialysis: *n* = 8	4.0 < dose ≤5.0 mg/kg: *n* = 7 5.0 < dose ≤6.0 mg/kg: *n* = 19 6.0 < dose ≤7.0 mg/kg: *n* = 17 ≤7.0 mg/kg: *n* = 4	HPLC-PDA	-Response at the lowest concentration (3.5 μg/mL) was significantly more than 5 times higher than that of the blank serum -Interday coefficient of variation for the lowest and highest concentration (200 μg/mL) samples was within 15%.	Cmax: 172.2–271.7 mg/L (median = 249.6; *n* = 6)	nd	[28]
**LZD**	Age: 78 Sex: male weight: 48.2 kg	Treatment of mediastinitis with TDM of serum and wound exudate concentrations of linezolid in renal function impaired patients.	Scr: 5.6 mg/dL glomerular filtration rate: 8.6 mL/min/1.73 m^2^	600 mg every 24 h After that, 300 mg every 24 h	HPLC	-Lower limit: 0.1 μg/mL -Intra/interday precision below 5.0%	Cmin: 11.5 μg/mL (Day 21) Cmin: 5.5 μg/mL (Day 55)	nd	[29]
	Age: 77 Sex: female weight: 55 kg	TDM was effective in preventing thrombocytopenia with linezolid: a case report	CLcr 29.9 mL/min	600 mg twice a day After that, 600 mg every 24 h	HPLC	-Lower limit: 0.25 μg/mL -Intra/interday precision below 5.0%	39.4 µg/mL (Day 9)	nd	[30]
	Age: 79 Sex: female weight: 58.5 kg	Successful combination therapy with linezolid and rifampicin with appropriate management of linezolid TDM in MRSA osteomyelitis: a case report	Scr 0.4 mg/dL	600 mg twice a day Thereafter, 300 mg twice a day At the time rifampicin is combined, 600 mg twice a day	HPLC	-Lower limit: 0.1 μg/mL -Intra/interday precision below 5.0%	Cmin: 15.1 µg/mL (Day 5) Cmin: 13.9 µg/mL (Day 8) As a result of combination therapy, Cmin was in the optimal range of 3.7 to 7.2 mg/mL.	nd	[31]
**TZD**	Population: *n* = 3 patients	Development of an assay system for simultaneous quantification of plasma concentrations of LZD, DAP, and TZD and its clinical application	CLcr 48.3–64.5 mL/min	200 mg once daily	UPLC-MS/MS	-TZD showed good linearity over wide ranges of 5–5000 ng/mL. -The lower limited of quantification and three quality controls (QCs: low, medium and high) were less than 15% for both accu-racy and precision. -Recovery rate of TZD: more than 84.8%	Cpeak and Cmin of TZD ranged from 1.87 to 4.92 μg/mL and from 0.09 to 0.78 μg/mL	nd	[32]
	Population: *n* = 3 patients	Development of an assay for simultaneous quantification of 12 antimicrobial agents commonly used in ICU and its clinical application	CLcr 51.7–60.4 mL/min	200 mg once daily	UHPLC-MS/MS	-The concentration ranges of calibration curves for TZD was 0.01–5 μg/mL. -The measured concentrations in blanks were less than 20% of the peak response of the lower limited of quantification and less than 5% for internal standard	The ranges of Cmin and Cpeak in patients with CLcr of 51.7–60.4 mL/min were 0.06–0.12 and 2.67–4.01 μg/mL	nd	[33]
**LZD**	Age: 78 Sex: male weight: 48.2 kg	Treatment of mediastinitis with TDM of serum and wound exudate concentrations of linezolid in renal function impaired patients.	Scr: 5.6 mg/dL glomerular filtration rate: 8.6 mL/min/1.73 m^2^	600 mg every 24 h After that, 300 mg every 24 h	HPLC	-Lower limit: 0.1 μg/mL -Intra/interday precision below 5.0%	Cmin: 11.5 μg/mL (Day 21) Cmin: 5.5 μg/mL (Day 55)	nd	[29]
	Age: 77 Sex: female weight: 55 kg	TDM was effective in preventing thrombocytopenia with linezolid: a case report	CLcr 29.9 mL/min	600 mg twice a day After that, 600 mg every 24 h	HPLC	-Lower limit: 0.25 μg/mL -Intra/interday precision below 5.0%	39.4 µg/mL (Day 9)	nd	[30]
	Age: 79 Sex: female weight: 58.5 kg	Successful combination therapy with linezolid and rifampicin with appropriate management of linezolid TDM in MRSA osteomyelitis: a case report	Scr 0.4 mg/dL	600 mg twice a day Thereafter, 300 mg twice a day At the time rifampicin is combined, 600 mg twice a day	HPLC	-Lower limit: 0.1 μg/mL -Intra/interday precision below 5.0%	Cmin: 15.1 µg/mL (Day 5) Cmin: 13.9 µg/mL (Day 8) As a result of combination therapy, Cmin was in the optimal range of 3.7 to 7.2 mg/mL.	nd	[31]
**TZD**	Population: *n* = 3 patients	Development of an assay system for simultaneous quantification of plasma concentrations of LZD, DAP, and TZD and its clinical application	CLcr 48.3–64.5 mL/min	200 mg once daily	UPLC-MS/MS	-TZD showed good linearity over wide ranges of 5–5000 ng/mL. -The lower limited of quantification and three quality controls (QCs: low, medium and high) were less than 15% for both accu-racy and precision. -Recovery rate of TZD: more than 84.8%	Cpeak and Cmin of TZD ranged from 1.87 to 4.92 μg/mL and from 0.09 to 0.78 μg/mL	nd	[32]
	Population: *n* = 3 patients	Development of an assay for simultaneous quantification of 12 antimicrobial agents commonly used in ICU and its clinical application	CLcr 51.7–60.4 mL/min	200 mg once daily	UHPLC-MS/MS	-The concentration ranges of calibration curves for TZD was 0.01–5 μg/mL. -The measured concentrations in blanks were less than 20% of the peak response of the lower limited of quantification and less than 5% for internal standard	The ranges of Cmin and Cpeak in patients with CLcr of 51.7–60.4 mL/min were 0.06–0.12 and 2.67–4.01 μg/mL	nd	[33]

HPLC:high performance liquid chromatography; HPLC-UV:high performance liquid chromatography with ultraviolet detection; Scr:serum creatinine; CSF: cerebrospinal fluid; IQR:interquartile range; RSD: relative standard deviation; nd: not described; LC-MS/MS:Liquid Chromatograph-tandem Mass Spectrometer; UPLC-MS/MS:ultra-performance liquid chromatography coupled to tandem mass spectrometry; HPLC-PDA:high-performance liquid chromatography equipped with photodiode array; CLcr:creatinine clearance; MRSA: methicillin resistant Staphylococcus aureus; UHPLC-MS/MS:ultra-high-performance liquid chromatography coupled with tandem mass spectrometry.

**Table 2 ijerph-19-02516-t002:** Pharmacokinetic parameters and penetration of each tissue.

	Skin and Soft Tissue		Intrapulmonary
	Adipose Tissue	Muscle	ELT
AUC_0–12_	5.3	5.9	NR
AUC_0–24_	NR	NR	106.0
AUC_tissue_/AUC_plasma_	1.1	1.2	39.7

AUC, area under the concentration time curve; ELT, epithelial lining fluid; NR, not reported.

## Data Availability

All data are applicable in the paper.

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
