# Peer review of "Importance and Reality of TDM for Antibiotics Not Covered by Insurance in Japan"

_ijerph, 2022, doi:10.3390/ijerph19052516_

Round 1
Reviewer 1 Report
The authors summarize the importance of TDM of antibiotics, which is not commonly practiced in Japan. While it contains useful information, its selection criteria are not clear. The authors discuss ceftriaxone, daptomycin, linezolid, and tedizolid, but are there any other antimicrobials for which blood levels should be measured? For example, TDM of beta-lactam drugs other than ceftriaxone is important in the fight against bacterial resistance, and I believe that it has been reported. It would be good if the selection criteria for the antimicrobials discussed in this article were clarified or other antimicrobials were also discussed. The term "antimicrobials" should be changed to "antibiotics" since this article only discusses the TDM of antibiotics. In addition, the title does not reflect the contents, so it should be reconsidered. In the section describing the characteristics of each antibiotic, please add information on the status of TDM in other countries so that it can be compared with Japan.
Author Response
Thank you for your reviewing our article. We made some changes in our manuscript according to reviewer’s suggestions with yellow highlight. We think some revises enhanced the quality of our manuscript.
Reviewer 1:
- The authors summarize the importance of TDM of antibiotics, which is not commonly practiced in Japan. While it contains useful information, its selection criteria are not clear. The authors discuss ceftriaxone, daptomycin, linezolid, and tedizolid, but are there any other antimicrobials for which blood levels should be measured? For example, TDM of beta-lactam drugs other than ceftriaxone is important in the fight against bacterial resistance, and I believe that it has been reported. It would be good if the selection criteria for the antimicrobials discussed in this article were clarified or other antimicrobials were also discussed.
Response: Thank you for your comment. We have added criteria for the selection of antibiotics (L48-51).
- The term "antimicrobials" should be changed to "antibiotics" since this article only discusses the TDM of antibiotics.
Response: Thank you for your comment. The term "antimicrobials" has been changed to "antibiotics" (title, L46, L346).
- The title does not reflect the contents, so it should be reconsidered.
Response: Thank you for your comment. We have changed the title of this review to " Importance and Reality of TDM for Antibiotics not Covered by Insurance in Japan"
- In the section describing the characteristics of each antibiotic, please add information on the status of TDM in other countries so that it can be compared with Japan.
Response: Thank you for your comment. There were some drugs for which there were no reports on the status of TDM in other countries, so we added a new section titled "6. The states of TDM in other countries (L337-).
Reviewer 2 Report
The topic of this review fits well the scope of the journal. The reviewer feels it can be accepted after various revision.
(1) There are a lot of typing mistakes in the manuscript. The authors should correct them.
(2) In Japan, when the hospital / clinics run TDM, do they validate the bioanalytical method according to the well recognized guidelines such as FDA, EMA or Japanese regulator? If so, the work is very time consuming. Is it practical to do so for limited number of patients?
(3) How frequent will the TDM be? If only limited patients need such service, it will be very costly to run the standard bioanalytical method validation.
(4) How much in term of percentage does the TDM accounts for the cost of medication?
Author Response
Thank you for your reviewing our article. We made some changes in our manuscript according to reviewer’s suggestions with yellow highlight. We think some revises enhanced the quality of our manuscript.
Reviewer 2
- There are a lot of typing mistakes in the manuscript. The authors should correct them.
Response: Thank you for your comment. We apologize for any inaccuracies in the description. We have corrected typos, unified units (L23-24, 28, 30, 35, 37, 40-42, 46, 77, 87, 150, 153, 193, 221, 230-231, 233-236, 242, 244, 249, 252, 258-259, 290, 299, 308, 310, 311, 312, 315, 319-322, 328).
We have proofread English with this MDPI in advance, so you can check it in the attached file.
- In Japan, when the hospital / clinics run TDM, do they validate the bioanalytical method according to the well recognized guidelines such as FDA, EMA or Japanese regulator? If so, the work is very time consuming. Is it practical to do so for limited number of patients?
Response: Thank you for your comment. As far as we know, some facilities follow the guidelines of the FDA, EMA, and Japanese regulatory authorities, but not all. As the number of drugs covered by insurance increases, there is a need to establish simpler and faster bioanalytical methods. We have added a new section "7. Limitation" to describe what you have pointed out(L347-353).
- How frequent will the TDM be? If only limited patients need such service, it will be very costly to run the standard bioanalytical method validation.
Response: Thank you for your comment. In Japan, many facilities lack access to timely bioanalytical method, and most facilities, including universities and laboratories, use research funds to conduct them. We have added a new section "7. Limitation" to describe what you have pointed out(L353-357).
- How much in term of percentage does the TDM accounts for the cost of medication?
Response: Thank you for your comment. As far as we could ascertain, we did not know the actual percentages.
Reviewer 3 Report
This article reviewed the therapeutic drug monitoring (TDM) of several antimicrobial agents including ceftriaxone, linezolid, daptomycin and tedizolid that are not covered by insurance, and described the clinical significance of therapeutic drug monitoring (TDM) and the reported methods for measuring blood concentrations in Japan. Overall, this article is well-written. I just have two minor suggestions.
- Please add more “clinical implication” following the “reports in Japan”.
- Like table 1 for ceftriaxone, please add three more tables to briefly summarize the report in Japan for each antibiotic.
Author Response
Thank you for your reviewing our article. We made some changes in our manuscript according to reviewer’s suggestions with yellow highlight. We think some revises enhanced the quality of our manuscript.
Reviewer 3
- Please add more “clinical implication” following the “reports in Japan”.
Response: Thank you for your comment. We have added "clinical implication" following "reports in Japan" for each antibiotic (CTRX (L117-118), DAP (L205-209), LZD (L261-263), TZD (L335-336)).
- Like table 1 for ceftriaxone, please add three more tables to briefly summarize the report in Japan for each antibiotic.
Response: Thank you for your comment. We added the contents of DAP, LZD, and TZD to Table 1. There were no reports of CSF concentrations measured for these drugs. In addition, the title of Table 1 and several words in the text have been revised (L108-109,126).